# A Decision Tree Model Using Urine Inflammatory and Oxidative Stress Biomarkers for Predicting Lower Urinary Tract Dysfunction in Females

**DOI:** 10.3390/ijms252312857

**Published:** 2024-11-29

**Authors:** Yuan-Hong Jiang, Jia-Fong Jhang, Jen-Hung Wang, Ya-Hui Wu, Hann-Chorng Kuo

**Affiliations:** 1Department of Urology, Hualien Tzu Chi Hospital, Buddhist Tzu Chi Medical Foundation, Hualien 97002, Taiwan; redeemerhd@gmail.com (Y.-H.J.); alur1984@hotmail.com (J.-F.J.); ryoma499@gmail.com (Y.-H.W.); 2Department of Urology, School of Medicine, Tzu Chi University, Hualien 97002, Taiwan; 3Department of Medical Research, Hualien Tzu Chi Hospital, Buddhist Tzu Chi Medical Foundation, Hualien 97002, Taiwan; jenhungwang2011@gmail.com

**Keywords:** urine biomarker, lower urinary tract dysfunction, decision tree model, detrusor overactivity, interstitial cystitis

## Abstract

Lower urinary tract dysfunction (LUTD) was associated with bladder inflammation and tissue hypoxia with oxidative stress. The objective of the present study was to investigate the profiles of urine inflammatory and oxidative stress biomarkers in females with LUTD and to develop a urine biomarker-based decision tree model for the prediction. Urine samples were collected from 31 female patients with detrusor overactivity (DO), 45 with dysfunctional voiding (DV), and 114 with bladder pain syndrome (BPS). The targeted analytes included 15 inflammatory cytokines and 3 oxidative stress biomarkers (8-hydroxy-2-deoxyguanosin, 8-isoprostane, and total antioxidant capacity [TAC]). Different female LUTD groups had distinct urine inflammatory and oxidative stress biomarker profiles, including IL-1β, IL-2, IL-8, IL-10, eotaxin, CXCL10, MIP-1β, RANTES, TNFα, VEGF, NGF, BDNF, 8-isoprostane, and TAC. The urine biomarker-based decision tree, using IL-8, IL-10, CXCL10, TNFα, NGF, and BDNF as nodes, demonstrated an overall accuracy rate of 85.3%. The DO, DV, and BPS accuracy rates were 74.2%, 73.3%, and 93.0%, respectively. Internal validation revealed a similar overall accuracy rate. Random forest models supported the significance and importance of all selected nodes in this decision tree model. The inter-individual variations and the presence of extreme values in urine biomarker levels were the limitations of this study. In conclusion, urine inflammatory and oxidative stress biomarker profiles of different female LUTDs were different. This internally validated urine biomarker-based decision tree model predicted different female LUTDs with high accuracy.

## 1. Introduction

Lower urinary tract symptoms (LUTS) are common and bothersome in both men and women [1,2]. The pathophysiology of female LUTS is complex and may involve bladder, urethral, and pelvic floor dysfunctions. However, clinical symptoms are unreliable in the diagnosis of lower urinary tract dysfunction (LUTD) in women [3]. Videourodynamic studies (VUDS) could provide a deep understanding of LUTS and accurately diagnose LUTD in female patients [4,5]. Due to the nature of invasive procedures, VUDS should be the second-line investigation after initial diagnosis and treatment based on symptoms or noninvasive tests fail. A comprehensive systemic review of the biomarkers associated with LUTS was recently published [6]. It analyzed multiple candidate biological pathways and suggested future research into biomarker-based precision medicine, which has so far remained elusive.

Bladder urothelial dysfunction and increased suburothelial inflammation were investigated in a variety of LUTDs, including interstitial cystitis/bladder pain syndrome (IC/BPS) [7], overactive bladder (OAB) [8], bladder outlet obstruction, and various bladder dysfunctions [9], which appeared to reflect their underlying pathophysiology. Recently, a significant association between oxidative stress and LUTS was reported [10]. The levels of advanced glycation end products and 8-hydroxy-2-deoxyguanosine (8-OHdG) in the urine were higher in severe LUTS patients. Controlling oxidative stress was thought to be a new therapeutic strategy for treating chronic ischemia-induced bladder dysfunction [11]. Both bladder inflammation and tissue hypoxia with oxidative stress were common pathophysiological bladder features in LUTD, and the roles of urine inflammatory and oxidative stress biomarkers in these LUTD were gradually revealed [6,12,13].

The OAB and IC/BPS in female patients commonly share indistinct symptoms that confuse clinical physicians, although their core symptoms for diagnosis are different. According to one recent research, the urine cytokine profiles of European Society for the Study of Interstitial Cystitis (ESSIC) type 2 IC/BPS and OAB patients differed significantly from those of controls [14]. Furthermore, a novel pilot diagnostic algorithm, identifying IC/BPS and OAB patients from controls, was developed with acceptable diagnostic rates. Another research found that female patients with dysfunctional voiding (DV) had significantly higher urine 8-OHdG, IL-1β, IL-8, and brain-derived neurotrophic factor (BDNF) levels than controls [15]. Based on these findings, the clinical application of urine biomarkers in females with LUTD is expected in the future.

We hypothesized that different LUTDs would have distinct protein profiles and biochemical contents due to their different pathophysiologies and intrinsic bladder conditions. Bladder inflammation and tissue hypoxia with oxidative stress were significant pathophysiological bladder features in LUTDs. This study investigated urine inflammatory and oxidative stress biomarker profiles in females with LUTD, including detrusor overactivity (DO), DV, and bladder pain syndrome (BPS), and developed a urine biomarker-based decision tree model for prediction.

## 2. Results

The clinical characteristics and VUDS parameters of DO, DV, and BPS patients are shown in Table 1. The DO patients were significantly older than DV and BPS patients (63.9 ± 9.0 vs. 53.2 ± 14.2, 54.6 ± 12.4, *p* < 0.001). The mean overactive bladder symptom score in DO patients was 9.4 ± 3.2. In DO and DV patients, the International Prostate Symptom Scores were 11.4 ± 5.8 and 15.1 ± 9.9, respectively. In the DV group, 53.3% (*n* = 24) of the patients had concomitant DO. In BPS patients, the mean O’Leary–Saint symptom score was 20.6 ± 7.9, with a mean visual analog scale of pain (VAS) of 4.2 ± 2.6 and a maximal bladder capacity under anesthesia of 717.8 ± 179.1 mL. Patients in all three groups had distinct VUDS parameters.

The urine biomarker profiles among different female LUTD groups are shown in Table 2. For each targeted analyte, the number of outliers within groups ranged from 0 to 1 in DO patients, from 0 to 2 in DV patients, and from 0 to 4 in BPS patients. Different female LUTD groups had significantly different urine inflammatory and oxidative stress biomarker profiles, including IL-1β, IL-2, IL-8, IL-10, eotaxin, chemokine (C-X-C motif) ligand 10 (CXCL10), macrophage inflammatory protein-1β (MIP-1β), regulated upon activation, normal T cell expressed and presumably secreted (RANTES), tumor necrosis factor α (TNFα), vascular endothelial growth factor (VEGF), nerve growth factor (NGF), BDNF, 8-isoprostane, and total antioxidant capacity (TAC).

Based on the significant differences in urine biomarker profiles among female LUTD groups, a urine biomarker-based decision tree model was built using the levels of IL-10 (root node, cutoff value 1.16 pg/mL), TNFα (cutoff value 1.08 pg/mL), BDNF (cutoff value 0.83 pg/mL), IL-8 (cutoff value 23.04 pg/mL), CXCL10 (cutoff value 22.82 pg/mL), and NGF (cutoff value 0.27 pg/mL) (Figure 1). This model demonstrated an overall accuracy rate of 85.3%, with individual accuracy rates of 74.2%, 73.3%, and 93.0% for DO, DV, and BPS, respectively.

Internal validation using the bootstrap method revealed that the accuracy rates of the model were 84.32% (95% CI 76.00, 92.50) and 85.40% (95% CI 76.81, 92.99) from the sampling data of the entire study patients, including and excluding outliers, respectively (Figure 2). The significance and importance of all selected nodes in this decision tree model were supported by the random forest models (Figure 3).

## 3. Discussion

To our knowledge, this is the first study to demonstrate an internally validated urine biomarker-based decision tree model for predicting female LUTD, including DO, DV, and BPS. Bladder inflammation and tissue hypoxia with oxidative stress were considered as important and common pathophysiological bladder features in different LUTD in females. However, the critical differences among these diseases were not clearly established. This study revealed that the urine inflammatory and oxidative stress biomarker profiles of these diseases differed. We developed a urine biomarker decision tree model that predicts the different LUTD in females with high accuracy, as supported by the internal validation results. Developing a urine biomarker-based decision tree model is a first step toward translating the urine biomarker profiles into clinical practice. It is a significant step forward in the advancement of precision medicine in LUTD.

With an overall accuracy rate of 85.3%, our urine biomarker-based decision tree model selected IL-8, L-10, TNFα, CXCL10, BDNF, and NGF as the nodes to diagnose the different LUTD in females, including DO, DV, and BPS. The IL-8 is a chemoattractant of neutrophils and T cells [16], that can regulate angiogenesis by directly enhancing the survival and proliferation of endothelial cells [17]. In redox signaling and oxidative stress, both IL-8 and TNFα are important released pro-inflammatory cytokines [18]. In a rat study of chronic bladder ischemia, both IL-8 and TNFα levels in the bladder tissue were elevated [19]. Neurotrophins, such as NGF and BDNF, play a role in neural control and sensory function in the urinary bladder and are known as biomarkers in both OAB and IC/BPS [20]. The higher the value of the mean decrease in accuracy or mean decrease in the Gini index, the higher the importance of the variable in the model. The data from random forest models demonstrated that all the selected nodes in this decision tree model were significant and important.

In this study, urine biomarker profiles of DO, DV, and BPS were distinct, but it was difficult to distinguish these groups by the analysis of single analyte. The previously reported diagnostic algorithm has acceptable diagnostic rates of 68.4%, 73.3%, and 60%, respectively, for distinguishing controls, ESSIC type 2 IC/BPS, and OAB patients [14]. The decision tree model is a very popular machine learning algorithm. A decision tree algorithm can be used to solve both regression and classification problems, and it has the advantages of being easy to understand, interpret, and visualize.

Despite DV being a bladder outlet obstruction condition, concomitant DO is common in such conditions. In our DV study population, 53.3% of patients also had concomitant DO, which may influence the design of the decision tree model nodes, cutoff values, and overall accuracy.

This urine biomarker-based decision tree, which uses urine levels of inflammatory and hypoxia-related cytokines, as well as neurotrophins, may provide a superior overall accuracy rate of 85.3% in diagnosing DO, DV, and BPS in female patients. The internal validation results revealed similarly high accuracy rates, demonstrating the discrimination and consistency of our prediction model. After importing more data, the decision tree model for predicting LUTD will be more accurate and reliable, pathophysiological, and applicable.

This study had several limitations. First, this urine biomarker-based decision tree model was developed using existing data from urine biomarker profiles at our institution. Although this decision tree model was internally validated, it will require external validation in the future. Second, all the enrolled female LUTD patients were medically refractory, and the accuracy rate may drop or differ when this model is applied to the female general population with LUTS. Moreover, a more comprehensive model is needed. Third, the shortcomings of urine biomarkers included intra-individual variations and the presence of extreme values. The extreme values in each study group were less than 5%, and we developed this decision tree with the exclusion of the extreme values. Under internal validation, the accuracy rate slightly reduced from 85.40% to 84.32% when the data were sampled from the entire study patients with outliers excluded versus without outliers excluded. This suggested that this model was still reliable. Fourth, the status of bladder volume may or may not affect the expression levels of urine biomarkers, as there is currently no evidence to confirm this. We chose to collect urine at the “full bladder” condition to minimize the impact of this variable and to facilitate application in subsequent research and clinical settings.

## 4. Materials and Methods

### 4.1. Patients and Investigation of Clinical Characteristics

From February 2015 to March 2021, we enrolled 31 DO, 45 DV, and 114 BPS female patients at the Department of Urology of a single medical center. All patients received VUDS with the indication of refractory LUTS, and their respective LUTD confirmed. The diagnostic details of VUDS were interpreted according to the International Continence Society’s terminology [21]. The following parameters of VUDS were recorded: first sensation of bladder filling (FSF), cystometric bladder capacity (CBC), detrusor voiding pressure (Pdet), maximal urinary flow rate (Qmax), corrected maximal urinary flow rate (cQmax, defined as Qmax/√CBC), voided volume (Vol), post-void residual volume (PVR), and voiding efficacy (VE, defined as Vol/CBC).

The inclusion criteria for DO, DV, and BPS patients were similar to our previous studies [14,15]. The DO patients were medically refractory OAB patients with DO evidence in VUDS. Patients diagnosed with DV had an open bladder neck, but a narrow membranous urethra or pelvic floor muscle level on real-time fluoroscopy, increased external urethral sphincter electromyography activities, and a low Qmax during voiding, without a history of neurological disease. The diagnostic criteria for BPS are based on the proposed guidelines of the European Society for the Study of Interstitial Cystitis [22]. All enrolled BPS patients, who received cystoscopic hydrodistention, and were essentially ESSIC type 1 or 2 IC/BPS (i.e., without or with glomerulations detected during hydrodistention). The grading of glomerulations was determined by the severity observed during the examination of five bladder regions (anterior, posterior, left lateral, right lateral, and bottom). The grades were defined as follows: normal (0), petechiae present in at least two quadrants (I), large areas of submucosal bleeding (II), diffuse and widespread submucosal bleeding (III), and mucosal disruption with or without associated bleeding (IV).

Exclusion criteria of enrolled patients included active urinary tract infection, neurogenic disorders (e.g., multiple sclerosis, spinal cord injury, cerebrovascular accidents, and Parkinson’s disease), and history of bladder surgery/or traumatic injury, urinary tract malignancy or tuberculosis, pelvic radiation, or nephrotic or nephritic syndrome, urolithiasis, and impaired renal function (serum creatinine > 2.0 mg/dL).

### 4.2. Assessment of Urine Biomarker Levels

Midstream urine samples were collected from all enrolled study patients. Urine was self-voided when subjects reported feeling full in the bladder. Before storing the urine samples, we performed urinalysis simultaneously to confirm their infection-free status, defined as having WBC < 10/HPF. In total, 50 mL of urine were immediately placed on ice and transferred to the laboratory for preparation. The samples were centrifuged at 1800 rpm for 10 min at 4 °C. Moreover, the supernatants were separated into aliquots in 1.5 mL tubes (1 mL per tube) and stored at −80 °C. Before further analysis, the frozen urine samples were centrifuged at 12,000 rpm for 20 min at 4 °C, and the supernatants were used for subsequent measurements.

### 4.3. Quantification of Inflammatory Cytokines

The inflammatory cytokines, chemokines, and neurotrophins investigated in urine samples were similar to those investigated in our previous study [23]. The targeted analytes in urine were assayed using commercially available microspheres with the Milliplex^®^ Human Cytokine/Chemokine magnetic bead-based panel kit (Millipore, Darmstadt, Germany). A total of 14 targeted analytes included IL-1β, IL-2, IL6, IL-8, IL-10, eotaxin, CXCL10, macrophage chemoattractant protein-1 (MCP-1), MIP-1β, RANTES, TNFα, and VEGF measured with the multiplex kit catalog number HCYTMAG-60K-PX30, NGF measured with the multiplex kit catalog number HADK2MAG-61K, and BDNF measured with the multiplex kit catalog number HNDG3MAG-36K. The following laboratory procedures of the quantification of these targeted analytes were performed similarly to those in our previous study [23].

### 4.4. Quantification of Prostaglandin E2 (PGE2)

The level of urine PGE2 was measured using a high-sensitivity ELISA kit (Cayman, Ann Arbor, MI, USA) according to the manufacturer’s instructions. The detailed procedures followed those reported in a previous study [24].

### 4.5. Quantification of Oxidative Stress Biomarkers

The measurements of 8-OHdG, 8-isoprostane, and TAC in urine samples were carried out in accordance with the manufacturer’s instructions (8-OHdG ELISA kit, Biovision, Waltham, MA, USA; 8-isoprostane ELIZA kit, Enzo, Farmingdale, NY, USA; Total Antioxidant Capacity Assay Kit, Abcam, Cambridge, MA, USA). The laboratory procedures followed those reported in a previous study [15].

The study was approved by the Institutional Review Board and Ethics Committee of Buddhist Tzu Chi General Hospital (No. IRB107-175-A and No. IRB107-37-A). All procedures were carried out in accordance with the relevant guidelines and regulations. We informed all study patients about the risks, rationale, procedures, ethics, and costs of this study, and all of them provided informed consent.

### 4.6. Statistical Analysis

Continuous variables were represented by means ± standard deviations, while categorical variables were represented by numbers and percentages. Outliers were defined as values that were outside the range between the means ± three standard deviations for each biomarker in each study group, and they were excluded from the development of the decision tree model. Mean differences in clinical data, as well as the levels of urine biomarkers, among groups were analyzed using one-way analysis of variance, and post hoc test was performed via Bonferroni’s correction. All calculations were performed using SPSS Statistics for Windows, Version 20.0 (IBM Corp., Armonk, NY, USA). If the *p*-value is less than 0.05, the difference is considered statistically significant.

### 4.7. Establishment of the Decision Tree Model

Following data collection and importation into the software, all analyses were carried out in R language (version 3.5.2), primarily using the party (for decision trees) and randomForest (for random forest) packages, and the decision tree and random forest models were established [25,26]. The decision tree model was used to develop a predictive model of LUTD based on predictor variables (urine biomarkers), whereas the random forest model was adopted to evaluate the importance of each urine biomarker. Moreover, the efficacy of the decision tree model was evaluated using accuracy. The decision tree was established based on the urine biomarker profiles among female LUTD groups. Biomarkers showing significant differences in expression levels between groups were selected as candidate nodes for the decision tree. Among various decision tree models, we chose the model with a high accuracy rate (>80%).

### 4.8. Internal Validation of the Decision-Tree Model

Internal validation was performed using the bootstrap method to assess the accuracy and consistency of our prediction models [27,28,29]. Thirty percent of testing data from our study population were sampled and used to evaluate the accuracy of prediction models. This procedure was repeated 1000 times, with the results used to calculate an unbiased estimate of the model accuracy and a 95% confidence interval (CI).

## 5. Conclusions

Female DO, DV, and BPS patients had distinct urine inflammatory and oxidative stress biomarker profiles. We developed and internally validated a urine biomarker-based decision tree to identify females with medically refractory LUTDs, achieving an accuracy rate of 85.3%. In the future, as more patients are enrolled and more urine analytes with significance are selected, a more comprehensive and accurate decision tree to diagnose female LUTD is expected.

## Figures and Tables

**Figure 1 ijms-25-12857-f001:**
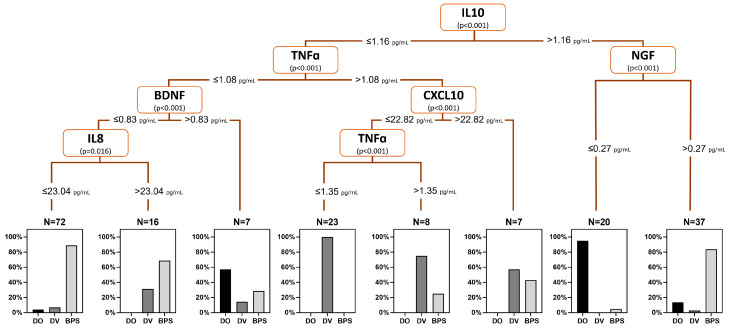
Urine biomarker-based decision tree model for the diagnosis for predicting female LUTD.

**Figure 2 ijms-25-12857-f002:**
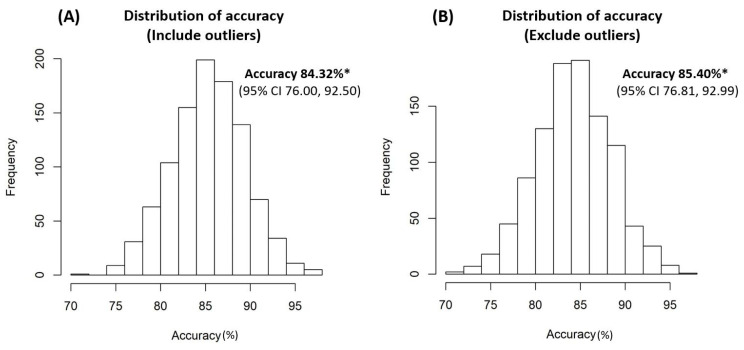
Internal validation results of the decision tree prediction model (**A**) including outliers, (**B**) excluding outliers. * Thirty percent of testing data from our study population (including or excluding outliers) were sampled using the bootstrap method. This procedure was repeated 1000 times, with the results used to calculate an unbiased estimate of the model accuracy and a 95% confidence interval (CI).

**Figure 3 ijms-25-12857-f003:**
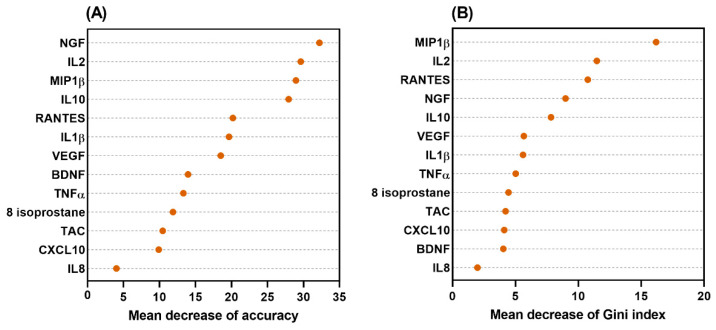
Results of the random forest models of the decision tree (**A**) the mean decrease of accuracy, (**B**) the mean decrease of Gini index.

**Table 1 ijms-25-12857-t001:** Clinical characteristics and VUDS parameters among different female LUTD groups.

	DO (*n* = 31)	DV (*n* = 45)	BPS (*n* = 114)	*p*-Value
Age	63.9 ± 9.0	53.2 ± 14.2	54.6 ± 12.4	<0.001
IPSS-V	3.9 ± 4.4	9.1 ± 7.2	NA	0.001
IPSS-S	7.5 ± 3.5	6.0 ± 4.0	NA	0.122
IPSS	11.4 ± 5.8	15.1 ± 9.9	NA	0.083
OABSS	9.4 ± 3.2	NA	NA	
OSS	NA	NA	20.6 ± 7.9	
VAS	NA	NA	4.2 ± 2.6	
MBC	NA	NA	717.8 ± 179.1	
Glomerulation grade	NA	NA	1.1 ± 0.9	
**VUDS**				
FSF	108.7 ± 48.6	125.2 ± 55.2	131.1 ± 59.9	0.124
CBC	286.3 ± 134.7	278.8 ± 134.3	251.6 ± 137.8	0.321
Pdet	18.0 ± 11.0	47.8 ± 42.7	22.3 ± 17.6	<0.001
Qmax	16.1 ± 7.3	10.6 ± 6.8	10.3 ± 6.4	<0.001
cQmax	1.0 ± 0.3	0.6 ± 0.4	0.6 ± 0.4	<0.001
Vol	272.1 ± 134	228.7 ± 115.8	211.9 ± 117.1	0.049
PVR	14.7 ± 40.8	56.4 ± 66	50.8 ± 102.8	0.093
VE	0.95 ± 0.11	0.80 ± 0.23	0.82 ± 0.27	0.017

VUDS, videourodynamic study; LUTD, lower urinary tract dysfunction; DO, detrusor overactivity; DV, dysfunctional voiding; BPS, bladder pain syndrome; IPSS, International Prostate Symptom Score; IPSS-S, International Prostate Symptom Score storage subscore; IPSS-V, International Prostate Symptom Score voiding subscore; OABSS, overactive bladder symptom score; OSS, O’Leary-Saint score; VAS, visual analog scale of pain; MBC, maximal bladder capacity under anesthesia; FSF, first sensation of bladder filling; CBC, cystometric bladder capacity; Pdet, detrusor voiding pressure; Qmax, maximal urinary flow rate; cQmax, corrected maximal urinary flow rate; Vol, voided volume; PVR, post-void residual volume; VE, voiding efficacy.

**Table 2 ijms-25-12857-t002:** Urine biomarker profiles among different female LUTD groups.

Urine Biomarkers *	1. DO *n* = 31	2. DV *n* = 45	3. BPS *n* = 114	*p*-Value	Post Hoc Analysis
IL-1β	0.61 ± 0.54 (1)	1.16 ± 1.4 (1)	0.64 ± 0.49 (3)	0.001	1, 3 < 2
IL-2	0.74 ± 0.19 (0)	0.28 ± 0.22 (0)	0.76 ± 0.18 (0)	<0.001	2 < 1, 3
IL-6	2.05 ± 2.62 (1)	2.14 ± 5.16 (2)	1.72 ± 1.53 (2)	0.674	
IL-8	20.67 ± 34.38 (1)	30.96 ± 63.85 (1)	14.17 ± 15.83 (2)	0.016	3 < 2
IL-10	1.54 ± 0.51 (1)	0.99 ± 0.11 (2)	1.07 ± 0.34 (2)	<0.001	2, 3 < 1
Eotaxin	6.04 ± 5.74 (1)	5.78 ± 7.3 (1)	8.62 ± 6.98 (3)	0.031	2 < 3
CXCL10	30.25 ± 45.83 (2)	10.64 ± 20.03 (1)	44.72 ± 58.44 (2)	<0.001	2 < 3
MCP-1	326.29 ± 304.61 (1)	196.09 ± 378.98 (1)	282.14 ± 242.36 (4)	0.128	
MIP-1β	3.66 ± 3.03 (1)	1.36 ± 4.02 (1)	3.16 ± 2.09 (3)	<0.001	2 < 1, 3
RANTES	8.81 ± 6.36 (0)	4.21 ± 7.82 (1)	9.33 ± 6.99 (2)	<0.001	2 < 1, 3
TNFα	0.87 ± 0.40 (1)	1.21 ± 0.33 (2)	0.78 ± 0.42 (2)	<0.001	1, 3 < 2
VEGF	14.63 ± 5.96 (0)	5.56 ± 4.91 (1)	14.41 ± 6.81 (1)	<0.001	2 < 1, 3
NGF	0.26 ± 0.07 (0)	0.21 ± 0.05 (1)	0.37 ± 0.17 (4)	<0.001	1, 2 < 3
BDNF	0.60 ± 0.22 (0)	0.63 ± 0.15 (0)	0.50 ± 0.17 (3)	<0.001	3 < 2
PGE2	261.77 ± 174.5 (0)	217.57 ± 186.73 (1)	239.40 ± 167.73 (3)	0.550	
8-isoprostane	32.53 ± 29.78 (0)	12.89 ± 14.70 (1)	39.09 ± 29.58 (2)	<0.001	2 < 1, 3
TAC	1558.76 ± 1358.87 (0)	604.35 ± 420.40 (2)	1657.94 ± 1189.74 (3)	<0.001	2 < 1, 3
8-OHdG	26.00 ± 17.68 (0)	32.42 ± 19.44 (0)	33.18 ± 17.92 (0)	0.150	

LUTD, lower urinary tract dysfunction; DO, detrusor overactivity; DV, dysfunctional voiding; BPS, bladder pain syndrome; IL, interleukin; CXCL10, chemokine (C-X-C motif) ligand 10; MCP-1, macrophage chemoattractant protein-1; MIP, macrophage inflammatory protein; RANTES, regulated upon activation, normal T cell expressed and presumably secreted; TNFα, tumor necrosis factor α; VEGF, vascular endothelial growth factor; NGF, nerve growth factor; BDNF, brain-derived neurotrophic factor; PGE2, prostaglandin E2; TAC, total antioxidant capacity; 8-OHdG, 8-hydroxy-2-deoxyguanosine; (): number of outliers; *: units: all pg/mL, except for ng/mL in 8-OHdG and mmol/μL in TAC.

## Data Availability

The data presented in this study are available on request from the corresponding author. The data are not publicly available due to restrictions imposed by the ethics committee of our institution.

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
