# Peer review of "A Decision Tree Model Using Urine Inflammatory and Oxidative Stress Biomarkers for Predicting Lower Urinary Tract Dysfunction in Females"

_ijms, 2024, doi:10.3390/ijms252312857_

Round 1
Reviewer 1 Report
Comments and Suggestions for Authors
This is an important report where the authors attempt to differentiate between three groups of female patients with LUTD (DO, DV and BPS) using urinary inflammatory and oxidate stress markers. This is a logical extension of earlier efforts from the same group to differentiate between OAB and BPS patients using urine biomarkers.
The authors developed and validated a decision tree that resulted in solid overall accuracy (85.3%) and impressive accuracy (93%) for determining BPS patients. The clinical implications of the study lie in the future ability to use non-invasive procedures to help diagnose female patients with LUTD>
The study is well designed and properly executed. There are some limitations that are freely acknowledged by the authors. Overall, this is an excellent study that awaits validation using a larger and more diverse sample pool and has the potential to significantly impact clinical decision making.
I have the following comments:
11. Did the authors examine accuracy rates for distinguishing between ESSIC type 1 vs type 2?
22. Did the authors intentionally omit BPS patients with Hunner ulcers? This is also an important group to consider during diagnosis.
33. Could the authors please expound their reasons for leaving outliers out of the original decision tree development? Upon validation, outliers did not result in significant loss of accuracy indicating their model is robust.
44. Table 1. I could not find how Glomerulation grade was calculated. Please explain in methods section.
55. In abstract, authors state “The intra-individual variations…”. From their description, only 1 sample was analyzed from each patient. Did the authors mean “inter-individual variations” or maybe just “individual variations”?
Author Response
Reviewer 1
This is an important report where the authors attempt to differentiate between three groups of female patients with LUTD (DO, DV and BPS) using urinary inflammatory and oxidate stress markers. This is a logical extension of earlier efforts from the same group to differentiate between OAB and BPS patients using urine biomarkers.
The authors developed and validated a decision tree that resulted in solid overall accuracy (85.3%) and impressive accuracy (93%) for determining BPS patients. The clinical implications of the study lie in the future ability to use non-invasive procedures to help diagnose female patients with LUTD.
The study is well designed and properly executed. There are some limitations that are freely acknowledged by the authors. Overall, this is an excellent study that awaits validation using a larger and more diverse sample pool and has the potential to significantly impact clinical decision making.
I have the following comments:
- Did the authors examine accuracy rates for distinguishing between ESSIC type 1 vs type 2?
Reply: Thank you for your insightful comment.
The enrolled BPS patients included both ESSIC type 1 and type 2 IC/BPS patients. As referenced in our study (Ref. 18: Am J Physiol Renal Physiol 2020, 318(6), F1391-F1399), there are some differences in the urine cytokine profiles between ESSIC type 1 and type 2 IC/BPS patients, though not all cytokines exhibit significant variation. Additionally, distinguishing between ESSIC type 1 and type 2 IC/BPS does not influence our clinical treatment strategy. Therefore, we did not specifically examine the accuracy rates for distinguishing between these subtypes in this study.
- Did the authors intentionally omit BPS patients with Hunner ulcers? This is also an important group to consider during diagnosis.
Reply: Thank you for your insightful comment, and we completely agree with your observation. BPS patients with Hunner’s ulcers typically exhibit significantly extremely higher levels of urine cytokines compared to those with non-Hunner IC, as reported in previous studies. Additionally, cystoscopy is considered the gold standard for identifying Hunner’s lesions. Given these distinct characteristics and the diagnostic reliance on cystoscopy, we did not include BPS patients with Hunner’s lesions in this study.
- Could the authors please expound their reasons for leaving outliers out of the original decision tree development? Upon validation, outliers did not result in significant loss of accuracy indicating their model is robust.
Reply: Thank you for your thoughtful comment. Based on previous studies investigating urine biomarker profiles (Refs. 14, 15, and 18), outliers accounting for approximately 1-3% of the data were excluded, as the analyses in those studies were conducted with outlier exclusion. Consequently, we developed the original decision tree model using data that excluded outliers to maintain consistency with prior methodologies. Importantly, during the validation process, we found that the inclusion or exclusion of outliers did not significantly impact the accuracy of the model, demonstrating its robustness.
- Table 1. I could not find how Glomerulation grade was calculated. Please explain in methods section.
Reply: Thank you for your comment. We have added the following description to the Methods section for clarification:
“The grading of glomerulations was determined by the severity observed during the examination of five bladder regions (anterior, posterior, left lateral, right lateral, and bottom). The grades were defined as follows: normal (0), petechiae present in at least two quadrants (I), large areas of submucosal bleeding (II), diffuse and widespread submucosal bleeding (III), and mucosal disruption with or without associated bleeding (IV).” (Page 3, Line 97-102)
- In abstract, authors state “The intra-individual variations…”. From their description, only 1 sample was analyzed from each patient. Did the authors mean “inter-individual variations” or maybe just “individual variations”?
Reply: Thank you for your comment. We have corrected "intra-individual variations" to "inter-individual variations" in abstract. (Page 1, Line 24)
Reviewer 2 Report
Comments and Suggestions for Authors
The topic of the article is very interesting and with potential to impact in a significant way the practical approach of female patients with lower urinary tract dysfunction.
This is not a new idea, and various articles were published in this regards. The merit of the paper resides in the evaluation of a decision tree, to create a logical and strategical approach when trying to diagnose these dysfunctions.
References 14 and 15 do not provide any significant informations to the reader. Taking into consideration that the inclusion criteria are listed following the statement "The inclusion criteria for DO, DV, and BPS patients were similar to our previous studies.” I believe that the above mentioned references should be removed.
Please explain where mentioning that the urine samples were collected when patients felt sensation of full bladder, the importance of this variable and how it may influence the results.
Please explain with more details how the decision tree was established. Also please detail why not other nodes were also included.
Regarding the conclusions, I think it should be stated again as clear as possible that the decision tree was used (and internally validated) to identify females with medically refractory lower urinary tract dysfunctions
Author Response
Reviewer 2
The topic of the article is very interesting and with potential to impact in a significant way the practical approach of female patients with lower urinary tract dysfunction.
This is not a new idea, and various articles were published in this regards. The merit of the paper resides in the evaluation of a decision tree, to create a logical and strategical approach when trying to diagnose these dysfunctions.
References 14 and 15 do not provide any significant informations to the reader. Taking into consideration that the inclusion criteria are listed following the statement "The inclusion criteria for DO, DV, and BPS patients were similar to our previous studies.” I believe that the above mentioned references should be removed.
Reply: Thank you for your comment. References 14 and 15 are foundational to our study and provide important context for the research.
Reference 14 highlights the development of a novel pilot diagnostic algorithm to differentiate OAB, IC/PBS, and controls based on the distinct urine cytokine profiles.
Reference 15 demonstrates that DV patients have significantly higher levels of urine 8-OHdG, IL-1β, IL-8, and brain-derived neurotrophic factor (BDNF) compared to controls.
These findings form the theoretical basis for our current study and are mentioned in the Introduction (Page 2, Lines 57–60 and Lines 65–67) and Discussion (Page 8, Line 270-272). We kindly request the reviewer to reconsider the inclusion of these references.
Please explain where mentioning that the urine samples were collected when patients felt sensation of full bladder, the importance of this variable and how it may influence the results.
Reply: Thank you for your comment. The effect of bladder volume on the expression levels of urine biomarkers remains unclear, as no definitive evidence currently exists to confirm its impact. To standardize the sampling process and minimize potential variability, we chose to collect urine samples when patients reported a sensation of a "full bladder." This approach was adopted to ensure consistency and to facilitate application in future research and clinical settings. The rationale for this timing aligns with the methodology used in References 14, 15, and 18. Additionally, we have addressed this limitation in the Discussion section (Page 8, Lines 301–303).
Please explain with more details how the decision tree was established. Also please detail why not other nodes were also included.
Reply: Thank you for your comment. The decision tree was established based on the urine biomarker profiles among female LUTD groups. Biomarkers showing significant differences in expression levels between groups were selected as candidate nodes for the decision tree. Among various decision tree models, we chose the model with a high accuracy rate (>80%). (We have added these sentences in Methods section, Page 4, Line 169-173).
Furthermore, the significance and importance of all selected nodes in this decision tree model were validated and supported by the results of the random forest analysis. (See Page 6, Line 231-232)
Regarding the conclusions, I think it should be stated again as clear as possible that the decision tree was used (and internally validated) to identify females with medically refractory lower urinary tract dysfunctions
Reply: Thank you for your comment. We have revised the conclusion for clarity. The updated description now reads:
“We developed and internally validated a urine biomarker-based decision tree to identify females with medically refractory LUTDs, achieving an accuracy rate of 85.3%.” (Page 8–9, Lines 309–311)